# Algorithm Design for Online Meta-Learning with Task Boundary Detection

Daouda Sow[1], Sen Lin[2], Yingbin Liang[1], Junshan Zhang[3]
[1]Department of ECE,  The Ohio State University
[2]Department of CS,  University of Houston
[3]Department of ECE,  University of California, Davis
sow.53@osu.edu, slin50@central.uh.edu, liang889@osu.edu, jazh@ucdavis.edu

Online meta-learning has recently emerged as a marriage between batch meta-learning and online learning, for achieving the capability of quick adaptation on new tasks in a lifelong manner. However, most existing approaches focus on the restrictive setting where the distribution of the online tasks remains fixed with known task boundaries. In this work, we relax these assumptions and propose a novel algorithm for task-agnostic online meta-learning in non-stationary environments. More specifically, we first propose two simple but effective detection mechanisms of task switches and distribution shift based on empirical observations, which serve as a key building block for more elegant online model updates in our algorithm: the task switch detection mechanism allows reusing of the best model available for the current task at hand, and the distribution shift detection mechanism differentiates the meta model update in order to preserve the knowledge for in-distribution tasks and quickly learn the new knowledge for out-of-distribution tasks. In particular, our online meta model updates are based only on the current data, which eliminates the need of storing previous data as required in most existing methods. We further show that a sublinear task-averaged regret can be achieved for our algorithm under mild conditions. Empirical studies on three different benchmarks clearly demonstrate the significant advantage of our algorithm over related baseline approaches.

## 1. Introduction

Two key aspects of human intelligence are the abilities to quickly learn complex tasks and continually update their knowledge base for faster learning of future tasks. Meta-learning [1–3] and online learning [4–6] are two main research directions that try to equip learning agents with these abilities. In particular, meta-learning aims to facilitate quick learning of new unseen tasks by building a prior over model parameters based on the knowledge of related tasks, whereas online learning deals with the problem where the task data is sequentially revealed to a learning agent. To achieve the capability of fast adaptation on new tasks in a lifelong manner, online meta-learning [3, 7, 8] has attracted much attention recently. Considering the setup where online tasks arrive one at a time, the objective of online meta-learning is to continuously update the meta prior based on which the new task can be learnt more quickly after the agent encounters more tasks.

In online meta-learning, the agent typically maintains two separate models, i.e., the *meta-model* to capture the underlying common knowledge across tasks and the *online task model* for solving the current task in hand. Most of the existing studies [3, 9] in online meta-learning follow a "resetting" strategy: quickly adapt the online task model from the meta model using the current data, update the meta model and reset the online task model back to the updated meta model at the beginning of the next task. This strategy generally works well when the task boundaries are known and the task distribution remains stationary. However, in many real-world data streams the task boundaries are not directly visible to the agent [7, 10, 11], and the task distributions can dynamically change during the online learning stage. Therefore, in this work we seek to solve the online meta-learning problem in such more realistic settings.

First Conference on Parsimony and Learning (CPAL 2024).

Needless to say, how to efficiently solve the online meta-learning problem without knowing the task boundaries in the non-stationary environments is nontrivial due to the following key questions: (1) *How to update the meta model and the online task model?* Clearly, the "resetting" strategy at the moment of new data arriving is not desirable, as adapting from the previous task model is preferred when the new data belongs to the same task with the previous data. On the other hand, the meta model update should be distinct between in-distribution (IND) tasks, where the current knowledge should be preserved, and out-of-distribution tasks (OOD), where the new knowledge should be learnt quickly. (2) *How to make the system lightweight for fast online learning?* The nature of online meta-learning precludes sophisticated learning algorithms, as the agent should be able to quickly adapt to different tasks typically without access to the previous data. And dealing with the environment non-stationarity should not significantly increase the computational cost, considering that the environment could change fast during online learning.

Our main contributions can be summarized as follows.

(1) (**Novel algorithm design**) We propose a novel online meta-learning algorithm in non-stationary environments without knowing the task boundaries, which appropriately addresses the problems above. More specifically, we first propose two simple but effective mechanisms to detect the task switches using the classification loss and detect the distribution shift using the Helmholtz free energy [12], respectively, as motivated by empirical observations. Based on these detection mechanisms, our algorithm provides a finer treatment on the online model updates, which brings in the following benefits: (1) (*task knowledge reuse*) The detection of task switches enables our algorithm to reuse the best model available for each task, avoiding the "resetting" to the meta model at each step as in most previous studies; (2) (*judicious meta model update*) The detection of distribution shift allows our algorithm to update the meta model in a way that the new knowledge can be quickly learnt for out-of-distribution tasks whereas the previous knowledge can be preserved for in-distribution tasks; (3) (*efficient memory usage*) Our algorithm does not reuse/store any of the previous data and updates the meta model at each online episode based only on the current data, which clearly differs from most existing studies [8, 10, 13] in online meta-learning.

(2) (**Extensive experiments**) We conduct extensive experiments in three different standard benchmarks for online meta-learning. As indicated by the experimental results, our algorithm significantly outperforms the related baselines methods on all benchmarks. The ablation study also verifies the effectiveness of the proposed detection mechanisms.

(3) (**Theoretical analysis**) We provide a regret analysis of the proposed algorithm by taking task boundary detection into account, where a sublinear task-averaged regret can be achieved under mild conditions. In particular, our result captures a trade-off between the impact of task similarity on the performance of standard online meta-learning with known task boundaries and the performance under task boundary detection uncertainty. Namely, when tasks are more similar, better performance can be achieved due to less task variations over time, but it is harder to detect task switches.

**Related Work: Meta-learning.** Also known as learning to learn, meta-learning [3, 14, 15] is a powerful tool for leveraging past experience from related tasks to quickly learn good task-specific models for new unseen tasks. As a pioneering method that drives recent success in meta-learning, model-agnostic meta-learning (MAML) [3] seeks to find good meta-initialization such that one or a few gradient descent steps from the meta-initialization leads to a good task-specific model for a new task. Several variants of MAML have been introduced [16–23]. Other approaches are essentially model based [2, 24–26] and metric space based [1, 14, 27, 28].

**Online Learning.** In online learning [4, 6, 29], cost functions are sequentially revealed to an agent which is required to select an action before seeing each cost. One of the most studied approach is follow the leader (FTL) [4], which updates the parameters at each step using all previously seen loss functions. Regularized versions of FTL have also been introduced to improve stability [30, 31]. Similar in spirit to our work in terms of computational resources, online gradient descent (OGD) [32] takes a gradient descent step at each round using only the revealed loss. However, traditional online learning methods do not efficiently leverage past experience and optimize for zero-shot performance

without any adaptation. In this work, we study the online meta-learning problem, in which the goal is to optimize for quick adaptation on future tasks as the agent continually sees more tasks.

**Online Meta-learning.** Online meta learning was first introduced in [13]. Pioneering methods [8, 13, 33] follow a FTL-like design approach, which requires storing previous tasks and leads to a linear growth of memory requirement. Follow-the-regularized-leader (FTRL) [31] approach has also been extended to the online meta learning setting in [34, 35], resulting in a better memory requirement. [9] proposed a memory-efficient approach based on summarizing previous task experiences into one state vector. However, these approaches require knowledge of task boundaries and "reset" the task model to the meta model at each online episode [3, 36]. Similar to [9], our algorithm also overcomes the linear memory scaling. But unlike their method, we do not need to know task boundaries and consider dynamic environments. [10] alleviates the "resetting" issue by updating the online model always starting from its previous state, which however needs to store previous models and has limited performance especially in dynamic environments where successive tasks can be very different.

None of the methods above considered the online meta-learning problem in a dynamic environment setting where the task distributions change substantially over time without knowing the task boundaries. [11] is the first work that empirically evaluated the proposed algorithm in a dynamic environment, but did not propose a method to quickly learn the knowledge for out-of-distribution tasks. In contrast, we update the meta representations in a way that preserves the in-distribution knowledge while continually improving fast adaptation for out-of-distribution tasks.

## 2. Background and Problem Formulation

**Background.** We first briefly introduce some related concepts about online meta-learning.

*Meta-learning via MAML.* Meta-learning [3, 14, 15], a.k.a., learning to learn, seeks to quickly learn a new task with limited samples by leveraging the knowledge from similar tasks. More specifically, the objective therein is to learn a meta model based on a set of tasks $\{\mathcal{T}_i\}_{i=1}^M$ drawn from some unknown distribution $\mathbb{P}(\mathcal{T})$, from which task-specific models can be quickly obtained for new tasks from the same stationary distribution $\mathbb{P}(\mathcal{T})$. Taking MAML as an example, the objective therein is to learn a model initialization $\theta$ such that one or a few gradient descent steps from $\theta$ can lead to a good model for a new task $\mathcal{T} \sim \mathbb{P}(\mathcal{T})$, by solving the following problem with training tasks $\{\mathcal{T}_i\}_{i=1}^M$:

$$\theta := \arg\min_\theta \frac{1}{M} \sum_{i=1}^M f_i\left(U_i(\theta)\right) \tag{1}$$

where task model $\phi_i = U_i(\theta) = \theta - \alpha\nabla\hat{f}_i(\theta)$, $\hat{f}_i$ and $f_i$ are the training and test losses for task $\mathcal{T}_i$.

*Online learning.* In the general online learning problem, loss functions are sequentially revealed to a learning agent: at each step $t$, the agent first selects an action $\theta_t$, and then a cost $f_t(\theta_t)$ is incurred. The goal of the agent is to select a sequence of actions to minimize the following *static* regret

$$\mathrm{R}(T) = \sum_{t=1}^T f_t\left(\theta_t\right) - \min_\theta \sum_{t=1}^T f_t\left(\theta\right), \tag{2}$$

i.e., the gap between the agent's predictions $\{f_t(\theta_t)\}_{t=1}^T$ and the performance of the best *static* model in hindsight. A successful agent achieves a regret $\mathrm{R}(T)$ that grows sublinearly in $T$. Online learning is a well studied field and we refer the readers to [37] for more information.

*Online meta-learning.* As a marriage between online learning and meta-learning, online meta-learning [7, 8, 13] aims to achieve the following two features: (i) fast adaptation to the current task (the meta-learning aspect); (ii) learn to adapt even faster as it sees more tasks (the online learning aspect). Specifically, the agent observes a stream of tasks $\mathcal{S} = \{\mathcal{T}_1, \mathcal{T}_2, ..., \mathcal{T}_T\}$ sampled from $\mathbb{P}(\mathcal{T})$, where tasks are revealed one at a time. For each task $\mathcal{T}_t$, the agent has access to a support set $S_t$ for task-specific adaptation and a query set $Q_t$ for evaluation. The goal here is to select a sequence of meta models $\{\mathbf{w}_t\}$ for achieving sublinear growth of the following regret

$$\mathrm{R}_{\mathrm{meta}}(T) = \sum_{t=1}^T f_t\left(U_t\left(\theta_t\right)\right) - \min_\theta \sum_{t=1}^T f_t\left(U_t(\theta)\right) \tag{3}$$

where $U_t$ is the task adaptation function depending on the support set $S_t$, and the cost function $f_t$ is evaluated using the adapted parameters $U_t(\mathbf{w}_t)$ on the query set $Q_t$. Intuitively, the agent seeks to learn a better meta model which leads to better task models for future tasks after seeing more tasks.

**Online meta-learning in non-stationary environments.** Differently from most online meta-learning studies [7–10, 13, 33], in this work we consider a more realistic scenario:

*Pre-trained meta model.* In many real applications, there is plenty of data available for pre-training, and it is unrealistic to deploy an agent in complex dynamic environments without any basic knowledge of the tasks at hand [11]. Therefore, following the same line as in [11], we assume that there is a set of training tasks $\{\mathcal{T}_i^0\}_{i=1}^M$ drawn from some unknown distribution $\mathbb{P}^0(\mathcal{T})$. And as standard in meta-learning, each pre-training task $\mathcal{T}_i^0$ has a support dataset $S_i^0$ and a query dataset $Q_i^0$. In this work, we employ MAML over the training tasks to learn a pre-trained meta model.

*Unknown task boundaries.* During the online meta-learning phase, we assume that the task boundaries are unknown, i.e., the so-called task-agnostic setup [11], in the sense that the agent does not know if the new coming data at time $t$ belongs to the previous task or a new task. To model the uncertainty about task boundaries, we assume that at any time $t$ the new data belongs to the previous task with probability $p \in (0, 1)$ or to a new task with probability $1 - p$.

*Non-stationary task distributions.* During the online meta-learning phase, the agent could encounter new tasks that are sampled from other distributions instead of the pre-training one $\mathbb{P}^0(\mathcal{T})$. To capture this non-stationarity in task distribution, we assume that whenever a new task arrives during online learning, it will be sampled either from $\mathbb{P}^0(\mathcal{T})$ with probability $\eta \in (0, 1)$ or from a new (w.r.t. $\mathbb{P}^0(\mathcal{T})$) distribution with probability $1 - \eta$. Note that we do not restrict the number of new distributions that can be encountered during online learning and the task distributions can be revisited.

## 3. Proposed Algorithm under Distribution Shifts

To address the online meta-learning problem mentioned above for non-stationary environments, we next propose a simple but effective algorithm, called onLine mEta lEarning under Distribution Shifts (LEEDS), based on the detection of task switches and distribution shift to assist fast online learning. Following most studies [10, 11] in online meta-learning, we maintain two separate models during the online learning stage: $\theta$ for the meta model and $\phi$ for the online task model.

**Detection of task switches and distribution shift:** To enable fast learning of a new task in online learning, the detection mechanisms can not be overly sophisticated, but have to be efficient with high detection accuracy. Towards this end, we propose two different methods for detecting the task switch and the distribution shift, respectively, which work in concert as key components of LEEDS.

*Detection of task switches.* To understand the learning behaviors under task switches, we evaluate the classification loss of the previous task model using the newly coming data, i.e., $\mathcal{L}(\phi_{t-1}; S_t)$ for time $t$, where $\mathcal{L}$ is the loss function, $\phi_{t-1}$ is the previous online model at time $t - 1$, and $S_t$ is the current support set. The left plot in Fig. 1 shows the empirical results on an online few-shot image recognition problem. As depicted, the loss value keeps decreasing as the agent receives more data from the same task but suddenly increases whenever a new task arrives. This is clearly reasonable as the learnt online model for the previous task does not fit the new task anymore. Inspired by this empirical observation, we use a simple mechanism based on the value of $\mathcal{L}(\phi_{t-1}; S_t)$ to detect the task boundaries: there is a task switch whenever the loss is above some pre-defined threshold. As demonstrated later in Section 5, such a simple mechanism is indeed quite effective as corroborated by its high detection accuracies on various online meta-learning problems.

*Detection of distribution shift.* To efficiently determine if a new task is IND or OOD, i.e., sampled from the pre-training task distribution or not, we consider an energy-based OOD detection mechanism with a binary classifier $C_\tau(\cdot; \theta)$ defined as follows

$$C_\tau(\mathbf{x}; \theta) = \begin{cases} 1 & \text{if } -\mathrm{E}(\mathbf{x}; \theta) \le \tau \\ 0 & \text{if } -\mathrm{E}(\mathbf{x}; \theta) > \tau \end{cases} \tag{4}$$

where $\mathrm{E}(\mathbf{x}; \theta) = -\delta \log \sum_{k=1}^K \exp\left(-g_k(\mathbf{x}; \theta)/\delta\right)$ corresponds to the Helmholtz free energy for input $\mathbf{x}$, $g_k(\cdot; \theta)$ is the $k$-th component of the prediction model's output, and $\delta$ is the temperature parameter. The hyperparameter $\tau$ is a threshold that can be set using the pre-training tasks. As shown in [12],

the negative energy is linearly aligned with the likelihood of the of input $\mathbf{x}$ under model $\theta$, making it a useful score for distinguishing IND and OOD tasks.

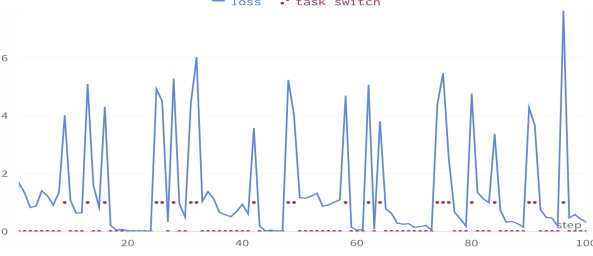

| Method | Memory |
|--------|--------|
| Ours | $\mathcal{O}(1)$ |
| [10] | $\mathcal{O}(T)$ |
| [11] | $\mathcal{O}(\frac{1}{1-p})$ |
| [9] | $\mathcal{O}(1)$ |
| [13] | $\mathcal{O}(T)$ |

Figure 1: **Left plot:** Variations of the online loss for a pre-trained meta model using MAML which is deployed for online learning. Red dot at 0 means no task switch at that time, and at 1 means the task switched at that time. **Right table:** Comparison of the memory requirements among different methods. $T$ is number of online rounds and $p \in (0, 1)$ is non-stationarity level.

**Update of meta and online parameters:** Based on the two detection schemes, the next question is how to update the meta and task models for fast adaption in dynamic environments.

Without such detection mechanisms, previous online meta-learning algorithms [9, 13] typically adapt the task model from the meta model using a support set, evaluate the adapted model on a query set, and reset the task model to the meta model when new data is received from the online data stream process, and then repeat the process again. However, such a "resetting" scheme can be sub-optimal in realistic scenarios. For instance, if the newly received data belongs to the same task with the previous data, the agent should update the task model by starting from previously adapted parameters instead of from the meta model.

In contrast, the simple but effective detection mechanisms in this work enable a more elegant treatment to the knowledge update during online learning:

(1) If there is a task switch at time $t$, i.e., $\mathcal{L}(\phi_{t-1}; S_t) > \ell$ where $\ell$ is the threshold, adapting from the meta model is generally better than adapting from the task model of the previous task. Therefore, we first obtain the online task model $\phi_t$ from the meta model using the new data:

$$\phi_t = \theta_{\text{adapt}} = \theta_{t-1} - \alpha_1 \nabla_{\theta_{t-1}} \mathcal{L}(\theta; S_t),$$

and then update the meta model no matter if there is a distribution shift, so as to incorporate the knowledge of the new task to the meta model:

$$\theta_t = \theta_{t-1} - \alpha_2 \nabla_\theta \mathcal{L}(\theta_{\text{adapt}}; Q_t).$$

(2) If there is no task switch, i.e., $\mathcal{L}(\phi_{t-1}; S_t) \leq \ell$, we continue to update the task model from the previous task model using the new data, different from the "resetting" scheme in the literature:

$$\phi_t = \phi_{t-1} - \alpha_1 \nabla_{\phi_{t-1}} \mathcal{L}(\phi_{t-1}; S_t).$$

To accelerate the knowledge learning for the new domains, we further distinguish the meta model update for IND and OOD tasks. In particular, if the current task is an IND task, we will only update the meta model once at the beginning of this task. That is to say, the meta model will not be further updated within the same task. In stark contrast, if the current task is an OOD task, we continue to update the meta model whenever new data for this task arrives as follows:

$$\theta_{\text{adapt}} = \theta_{t-1} - \alpha_1 \nabla_\theta \mathcal{L}(\theta_{t-1}; S_t), \ \ \theta_t = \theta_{t-1} - \alpha_2 \nabla_\theta \mathcal{L}(\theta_{\text{adapt}}; Q_t).$$

The details of LEEDS can be found in Algorithm 1. Note that the performance of LEEDS is indeed robust to the threshold parameters $\ell$ and $\tau$ as shown later in our experimental results, and we also explain the heuristics for setting the thresholds in a more effective manner in the appendix.

**Memory friendly:** One important feature of the proposed algorithm LEEDS is that the meta model update is only based on the current data. In contrast, most of the previous studies store the previous data in the memory for the meta model update. A comparison of the memory requirements among

**Algorithm 1** onLine mEta lEarning under Distribution Shifts (LEEDS)

---

1: **Input:** Dynamic stream $\mathcal{S}$, pre-training distribution $\mathbb{P}^0(\mathcal{T})$, stepsizes $\alpha_1$ and $\alpha_2$, thresholds $\ell$ and $\tau$.
2: Perform pre-training phase using MAML method on tasks drawn from $\mathbb{P}^0(\mathcal{T})$
3: **while** stream $\mathcal{S}$ is ON **do**
4:    $D_t \longleftarrow \mathcal{S}$ // receive current data from online data stream $S_t, Q_t$
5:    $S_t, Q_t \longleftarrow D_t$ // split data into support and query
6:    **if** $\mathcal{L}(\phi_{t-1}; S_t) \leq \ell$ (i.e., no switch) **then**
7:       $\phi_t = \phi_{t-1} - \alpha_1 \nabla_{\phi_{t-1}} \mathcal{L}(\phi_{t-1}; S_t)$ // adapt starting from previous online model
8:       Evaluate $\phi_t$ on query set $Q_t$
9:       **if** $C_\tau(S_t; \theta_{t-1})$ (i.e., covariate shift) **then**
10:          $\theta_{\text{adapt}} = \theta_{t-1} - \alpha_1 \nabla_\theta \mathcal{L}(\theta_{t-1}; S_t), \theta_t = \theta_{t-1} - \alpha_2 \nabla_\theta \mathcal{L}(\theta_{\text{adapt}}; Q_t)$ // update meta model
11:       **end if**
12:    **else**
13:       $\theta_{\text{adapt}} = \theta_{t-1} - \alpha_1 \nabla_{\theta_{t-1}} \mathcal{L}(\theta_{t-1}; S_t)$ // adapt starting from meta model using support set
14:       Set $\phi_t = \theta_{\text{adapt}}$ and Evaluate $\phi_t$ on query set $Q_t$
15:       $\theta_t = \theta_{t-1} - \alpha_2 \nabla_\theta \mathcal{L}(\theta_{\text{adapt}}; Q_t)$ // update meta model
16:    **end if**
17: **end while**

---

different approaches is summarized in the right table in Fig. 1. As will be shown later in the experiments, LEEDS significantly outperforms the related baselines without the need of storing any previous data. This encouraging result suggest that the community could also learn from careful explorations of simpler designs, besides emphasizing algorithmic complexities.

## 4. Theoretical Results

In the online meta-learning with distribution shifts, it is clear that the *static* comparator in Equation (3) is not sufficient to capture the non-stationarity, as one cannot expect to have a single meta-model for all task distributions. Hence, we consider a task-averaged regret (TAR) by following [34]

$$\mathbf{R}_{\text{avg}} = \frac{1}{T} \sum_{t=1}^{T} \left( \sum_{k=1}^{K_t} f_t^k(\phi_t^k) - \sum_{k=1}^{K_t} f_t^k(\phi_t^*) \right) \tag{5}$$

where $T$ is the number of tasks, $K_t$ is the number of steps within task $t$, and $\phi_t^*$ is the dynamic comparator for each task $t$. For simplicity, we denote $f_t^k$ as the within-task loss function in step $k$ of task $t$ (evaluated on the query set). As shown in [34], one cannot expect to achieve a TAR that decreases w.r.t. $T$ because the dynamic comparators $\{\phi_t^*\}_t$ can force a constant loss for each task $t$. However, the average is still not taken w.r.t. the total number of steps $\sum_{t=1}^{T} K_t$, but only w.r.t. the number of tasks $T$. Therefore, the objective here is to achieve a TAR that is sublinear in $K_t$. Note that the within-task loss functions $\{f_t^t\}_{k=1}^{K_t}$ are usually non-adversarial in many practical online meta-learning problems. For example, in few-shot online image classification problem, $\{f_t^t\}_{k=1}^{K_t}$ correspond to different evaluations of the same classification loss function on different query sets for the same task. In what follows, we can show that a constant TAR is achievable for the non-adversarial case even after taking the detection errors of task boundaries into consideration.

Let the comparator $\phi_t^*$ be a minimizer of each of the loss functions $f_t^k, k = 1, ..., K_t$. To formally characterize the non-adversariality of the within-task loss functions $\{f_t^t\}_{k=1}^{K_t}$ in online meta-learning, we make the following assumptions.

**Assumption 4.1.** The comparator $\phi_t^*$ is a fixed point of the adaptation mappings $U_t^k$ at step $k$ within task $t$, i.e., $U_t^k(\phi_t^*) = \phi_t^*$ for $k = 1, ..., K_t$.

**Assumption 4.2.** The adaptation mapping $U_t^k$ is a contraction, i.e., there exists some $\rho_t^k$ such that for all $\phi_1, \phi_2$, $\|U_t^k(\phi_1) - U_t^k(\phi_2)\| \leq \rho_t^k \|\phi_1 - \phi_2\|$.

Assumption 4.1 characterizes the property of the dynamic comparator $\phi_t^*$. It is clear that Assumption 4.1 will hold when there exists a task model $\phi$ that can perfectly achieve zero loss on all data points sampled from the data distribution for task $t$. For example, in over-parameterized regime,

one can always train a model to perfectly predict all training data points with near-zero loss. Assumption 4.2 can be easily satisfied in online meta-learning [13]. For example, the one step gradient descent mapping, i.e., $U_t^k(\phi) = \phi - \alpha\nabla\hat{f}_t^k(\phi)$, satisfies Assumption 4.2 when the function $\hat{f}_t^k$ is $\beta$-smooth and $\mu$-strongly convex, and the stepsize $\alpha$ is chosen in $(0, \frac{2}{\beta})$. Define $\rho = \max_{t,k} \rho_t^k$.

**Assumption 4.3.** Each function $f_t^k$ is $L$-smooth.

Note that we do not make any assumption on the convexity of the within-task loss functions $f_t^k$. For any step $r$ in the online meta-learning process (where each task $t$ includes $K_t$ steps), let $P_r$ be the current data distribution and $\phi_r$ be the task model obtained after adaptation. To analyze the error probability of the detection mechanisms proposed in this work, we make the following assumption on the single-data loss function $\ell$:

**Assumption 4.4.** Let $\ell(\cdot, \xi)$ be the loss on a data point $\xi$ and $0 \leq \ell(\cdot, \xi) \leq M$. We assume that there exist constants $\ell_m \leq \ell_p$ such that the following holds for any step $r$ and $s$: (1) $E_{\xi \sim P_r}\ell(\phi_r, \xi) \leq \ell_m$; (2) $E_{\xi \sim P_r}\ell(\phi_s, \xi) \geq \ell_p$ if $P_r \neq P_s$.

Assumption 4.4 characterizes the comparison of the expected loss w.r.t. a certain data distribution between 1) the model adapted on this distribution and 2) the model adapted on another distribution. This essentially means, in expectation, the loss after adaptation is less than the loss of another task model, which is crucial for a threshold-based detection scheme to be applicable.

By updating the meta-model using OGD [32] on the meta loss $\left\|\phi_t^0 - \phi_t^*\right\|^2$ after each task $t$, we can have the following theorem to characterize the expected TAR w.r.t. the uncertainty of the task-boundary detection.

**Theorem 4.5.** *Suppose Assumptions 4.1,4.2, 4.3, 4.4 hold. Let $R = \sum_{t=1}^T K_t$ be the total number of online rounds, and $S = c \log R$ be the number of data points used for adaptation at each step, where $c$ is some positive constant. Then the expected TAR is bounded as*

$$\mathbb{E}[\mathbf{R}_{\mathrm{avg}}] \leq \mathcal{O}\left(\frac{\sigma_*^2 + \frac{\log T}{T}}{1 - \rho^2} + R^{-\left(\frac{c(\ell_p - \ell_m)^2}{2M^2} - 2\right)}\right),$$

*where the expectation is taken over the task-boundary detection uncertainty. $\sigma_*^2 = \frac{1}{T}\sum_{t=1}^T \left\|\phi_t^* - \phi^*\right\|^2$ and $\phi^* = \frac{1}{T}\sum_{t=1}^T \phi_t^*$ denote the variance and the mean of the comparators $\{\phi_t^*\}_{t=1}^T$, respectively. In particular, a constant expected TAR can be achieved by selecting $c > \frac{4M^2}{(\ell_p - \ell_m)^2}$.*

Intuitively, the first term in the upper bound in Theorem 4.5 captures the TAR for online meta-learning with known task boundaries, whereas the second term characterizes the impact of the task-boundary detection uncertainty. As shown in Theorem 4.5, when the tasks become more similar, the first term will decrease because the variance $\sigma_*^2$ is smaller, whereas the second term can increase because the constants $\ell_p$ and $\ell_m$ become closer (i.e., it is harder to detect task switches if tasks are more similar). Therefore, Theorem 4.5 captures a trade-off between the impact of task similarity on the performance of standard online meta-learning and the performance under the task boundary detection uncertainty. In practice, the optimal actions $\phi_t^*$ are usually not available for the meta updates. Thus, in our algorithm we use the alternative MAML-like updates which has been shown to be effective for learning meta parameters. As demonstrated in the different experiments in the next section, such updates can indeed serve as a good practical alternative.

## 5. Experiments

**Experimental setup.** We investigate the performance of LEEDS on three standard online meta-learning benchmarks, Omniglot-MNIST-FashionMNIST, Tiered-ImageNet and Synbols, compared to multiple related baseline algorithms. Specifically, we pre-train the meta model in one domain and then deploy it in a dynamic environment where tasks can be drawn from new domains. We evaluate all the algorithms using the average of test losses obtained throughout the entire online learning stage. To investigate the impact of the non-stationary level on the learning performance, we further consider two different cases of the environment non-stationarity: A moderately stationary case where the probability of not switching to a new task is set to $p = 0.9$, and a low stationary case where $p = 0.75$. We do not consider the cases where $p$ is very small, as an algorithm that just assumes

task switches at each round should perform well in such cases. We compare algorithms over 10000 episodes unless otherwise stated. See more details about datasets and baselines in Appendix C. More comprehensive results and training curves of different methods are also provided in Appendix B.

For all the experiments, whenever a new task needs to be revealed, it will be drawn from either the pre-training domain with probability $0.5$, or from one of the OOD domains with probability $0.5$. For Tiered-ImageNet, because only ood2 is trully OOD w.r.t. the pre-training task distribution, we increase the sampling probability of ood2 to $0.5$ which is consistent to the protocol $50\% - 50\%$ for IND and OOD tasks in all our experiments. More details about the experimental setup including the neural network architectures and the hyperparameter search are deferred to Appendix D.

Table 1: Average accuracy over 10000 online episodes on **Omniglot-MNIST-FashionMNIST** benchmark under different non-stationarity levels. "pre-train" domain: **Omniglot**; "ood1" domain: **MNIST**, "ood2" domain: **FashionMNIST**. The advantage of our algorithm LEEDS over the other baselines is more significant in the ood domains. See Appendix C for more details about the datasets.

| Method | non-stationarity level $p = 0.9$ | | | non-stationarity level $p = 0.75$ | | |
|---|---|---|---|---|---|---|
| | pre-train | ood1 | ood2 | pre-train | ood1 | ood2 |
| LEEDS | **99.39** ±0.09 | **96.44** ±0.11 | **82.87** ±0.19 | **98.97** ±0.10 | **95.68** ±0.12 | **81.49** ±0.22 |
| CMAML++ | 98.78 ±0.12 | 92.52 ±0.19 | 76.16 ±0.28 | 97.39 ±0.11 | 89.07 ±0.20 | 73.35 ±0.35 |
| CMAML | 89.79 ±0.54 | 84.06 ±0.80 | 69.70 ±0.63 | 75.51 ±0.94 | 70.41 ±1.22 | 58.58 ±1.27 |
| FOML | 89.20 ±0.61 | 70.84 ±0.76 | 64.83 ±0.74 | 81.68 ±0.59 | 59.24 ±0.78 | 58.07 ±0.77 |
| MAML | 95.07 ±0.10 | 62.02 ±0.14 | 54.67 ±0.17 | 95.51 ±0.11 | 62.31 ±0.13 | 54.83 ±0.13 |
| ANIL | 96.54 ±0.12 | 42.14 ±0.16 | 40.12 ±0.13 | 96.88 ±0.11 | 42.08 ±0.11 | 40.00 ±0.13 |
| MetaOGD | 84.05 ±1.66 | 73.73 ±1.39 | 60.03 ±1.60 | 85.67 ±1.57 | 75.09 ±1.43 | 60.51 ±1.65 |
| BGD | 63.58 ±2.25 | 46.12 ±2.10 | 44.89 ±1.41 | 23.86 ±2.36 | 17.97 ±2.17 | 19.81 ±1.63 |
| MetaBGD | 77.73 ±1.26 | 59.11 ±0.84 | 54.67 ±0.82 | 43.95 ±1.31 | 27.87 ±0.94 | 30.14 ±0.98 |

## 5.1. Main results

**Results on Omniglot-MNIST-FashionMNIST (OMF).** The online evaluations of the compared methods are shown in Table 1 for non-stationary levels $p = 0.9$ and $p = 0.75$. For each setting we report separately the online accuracies on pre-training domain and on the other two OOD domains, to show how our method keeps improving on the OOD domains while also remembering the pre-tarining tasks. Clearly, our method LEEDS achieves superior performance compared to all other baseline algorithms in both settings. More specifically, on the IND domain all methods pre-trained using MAML perform similarly, but are outperformed by LEEDS and CMAML++ which can detect task boundaries. However, on the OOD domains our algorithm significantly outperforms all other baselines, including CMAML++. This is due to the key OOD adaptation module that allows LEEDS to dynamically adapt the meta model based on the task distribution. Interestingly, comparing the performance for MAML and ANIL provides some insights on the limitations of re-using pre-trained representations in non-stationary environments. In fact, the ANIL baseline, which does not adapt its inner representations, performs poorly compared to MAML on the OOD domains, but achieves similar results on the pre-training domain. Also, the results highlight some limitations of the recently introduced FOML [10] method, which achieves lower performance than other competitive baselines. This is because FOML requires the tasks to be not mutually exclusive, which may not hold for the standard few-shot benchmarks considered in our experiments.

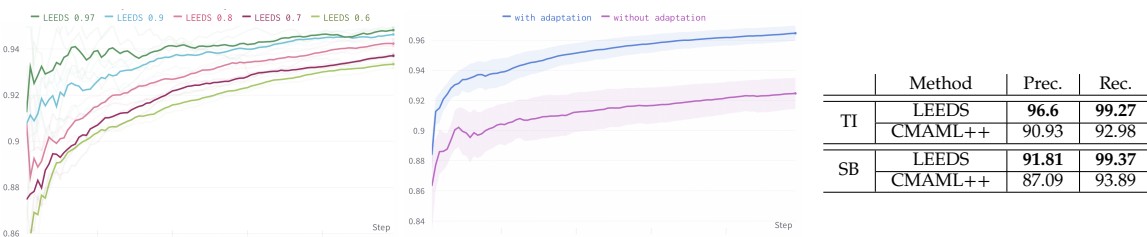

Figure 2: **Left:** LEEDS under different $p$. **Center:** LEEDS with and without domain adaptation. **Right:** Task boundaries detection on **Tiered-ImageNet (TI)** and **Synbols (SB)**.

**Results on Tiered-ImageNet (TI) and Synbols (SB).** We report the online accuracies on all domains and on OOD domains for these two benchmarks in Table 3. Because the distribution of the pre-

Table 2: Average accuracy over 20000 and 10000 online episodes on **Tiered-ImageNet** benchmark and **Synbols** benchmark, respectively, under different non-stationarity levels. The different domains are distinct splits of the original Tiered-ImageNet dataset (please see experimental setup in Appendix C for details on how these splits are obtained). In **Synbols**, the "pre-train" domain corresponds to 3 different alphabets from Synbols dataset, "ood1" corresponds to a new (w.r.t. the pre-training one) alphabet from Synbols dataset, and "ood2" contains font classification tasks. Full table with variance is in Appendix B.

| Method | Tiered-ImageNet | | | | Synbols | | | |
| --- | --- | --- | --- | --- | --- | --- | --- | --- |
| | non-stationarity $p = 0.9$ | | non-stationarity $p = 0.75$ | | non-stationarity $p = 0.9$ | | non-stationarity $p = 0.75$ | |
| | all | ood | all | ood | all | ood | all | ood |
| LEEDS | **66.07** | **67.43** | **64.52** | **65.80** | **85.12** | **67.48** | **82.22** | **63.68** |
| CMAML++ | 63.83 | 63.75 | 61.28 | 61.96 | 81.14 | 62.39 | 79.74 | 60.70 |
| FOML | 35.90 | 35.87 | 32.02 | 31.61 | 46.40 | 41.73 | 37.46 | 34.13 |
| MAML | 62.37 | 61.00 | 62.54 | 60.88 | 76.25 | 42.70 | 74.87 | 43.84 |
| ANIL | 59.78 | 57.61 | 59.57 | 57.38 | 64.58 | 34.51 | 72.69 | 35.66 |
| MetaOGD | 57.01 | 57.32 | 56.80 | 56.94 | 72.04 | 46.69 | 67.93 | 42.66 |
| BGD | 40.95 | 41.44 | 35.48 | 35.97 | 25.63 | 25.61 | 27.53 | 27.17 |
| MetaBGD | 49.21 | 50.01 | 44.58 | 45.30 | 53.74 | 42.25 | 40.79 | 34.63 |

training tasks is similar to the OOD ones for the Tiered-ImageNet benchmark, methods such as MAML can perform reasonably well. In fact, in the lower non-stationary case ($p = 0.75$), MAML is able to outperform the more complex CMAML++ baseline. However, our algorithm still achieves the best performance under both non-stationary levels and in both benchmarks. Note that in the larger **TI** dataset case, the FOML algorithm, which stores all previously seen tasks, runs out of memory after around $6500$ online episodes. Again because of similarity between OOD and IND tasks in the **TI** benchmark, static representations learned by ANIL are useful for all domains.

## 5.2. Ablation Studies

**Task boundaries detection.** The table in the right of Figure 2 provides the precision and recall scores of the task switch detection schemes for our method and CMAML++. Our detection scheme outperforms that of CMAML++ in all metrics. This is because, the detection scheme in CMAML++ is based on comparing successive losses, which could lead to over detection of task boundaries, especially when the task loss is too high at the first time the task is revealed to the online algorithm.

**Importance of domain adaptation module.** We investigate the importance of the distribution shift detection module that allows our algorithm LEEDS to update the meta model differently for in-distribution and out-of-distribution tasks. Fig. 2 shows the performance of our algorithm with and without the distribution shift detection module. The performance of the algorithm significantly improves ($\sim 4.3\%$ improvement) with this module. This shows that such a simple mechanism can effectively boost the online learning performance by allowing the agent to learn more from OOD data while also remembering pre-training knowledge.

**Sensitivity to frequency of task switches.** Fig. 2 shows the performance of our algorithm for different values of the probability $p$ of task switches. The performance increases with $p$, which shows that our algorithm LEEDS can successfully re-use previous task knowledge to increase performance.

## 6. Conclusions

In this work, we study the online meta-learning problem in non-stationary environments without knowing the task boundaries. To address the problems therein, we propose LEEDS for efficient meta model and online task model updates. In particular, based two simple but effective detection mechanisms of the task switches and the distribution shift, LEEDS can efficiently reuse the best task model available without resetting to the meta model and distinguish the meta model updates for in-distribution tasks and out-of-distribution tasks so as to quickly learn the new knowledge for new distributions while preserving the old knowledge of the pre-training distribution. In particular, the meta model update in LEEDS is based on the current data only, eliminating the need of storing previous data. Extensive experiments corroborate the superior performance of LEEDS over related baseline methods on multiple benchmarks.

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

# Supplementary Material

We provide the details omitted in the main paper. The sections are organized as fellows:

• Appendix B: We provide more empirical results including sensitivity to the thresholds and online evaluations curves for all settings.

• Appendix C: We provide further details about datasets and baseline methods.

• Appendix D: We provide further experimental specifications and discuss the heuristics used to set the thresholds.

• Appendix F: We provide our proof of Theorem 4.5.

## A. More Related Work about Continual Learning

**Continual Learning.** Continual learning (CL; a.k.a lifelong learning) focuses on overcoming "catastrophic forgetting" [38, 39] when learning from a sequence of non-stationary data distributions. Existing approaches are rehearsal-based [40–42], regularization-based [43, 44], and expansion-based [45–47]. For instance, rehearsal-based methods store a subset of previous tasks data and reuse it for experience "replay" to avoid forgetting. However, traditional CL methods evaluate the final model on all previously seen tasks so as to measure forgetting. In this work we are interested in online meta-learning and evaluate models with the average online performance (e.g., accuracy) after adaptation, which better captures the ability to quickly adapt to new online tasks [11]. Even though we do not specifically focus on avoiding forgetting, we update the meta-model in a way that preserves knowledge of in-distribution domain while also improving fast adaptation for out-of-distribution domains, as demonstrated in our various experiments.

## B. More Experimental Results

### B.1. Accuracy tables with variances for Tiered-ImageNet and Synbols datasets

| Method | non-stationarity $p = 0.9$ | | non-stationarity $p = 0.75$ | |
| --- | --- | --- | --- | --- |
| | all domains | ood domains | all domains | ood domains |
| LEEDS | **66.07** ±0.24 | **67.43** ±0.38 | **64.52** ±0.17 | **65.80** ±0.31 |
| CMAML++ | 63.83 ±0.27 | 63.75 ±0.55 | 61.28 ±0.23 | 61.96 ±0.41 |
| FOML | 35.90 ±0.56 | 35.87 ±0.83 | 32.02 ±0.42 | 31.61 ±0.69 |
| MAML | 62.37 ±0.46 | 61.00 ±0.72 | 62.54 ±0.37 | 60.88 ±0.65 |
| ANIL | 59.78 ±0.21 | 57.61 ±0.38 | 59.57 ±0.22 | 57.38 ±0.36 |
| MetaOGD | 57.01 ±0.28 | 57.32 ±0.66 | 56.80 ±0.25 | 56.94 ±0.42 |
| BGD | 40.95 ±0.85 | 41.44 ±1.15 | 35.48 ±0.76 | 35.97 ±1.09 |
| MetaBGD | 49.21 ±1.05 | 50.01 ±1.25 | 44.58 ±1.12 | 45.30 ±1.20 |

Table 3: Average accuracy over 20000 online episodes on **Tiered-ImageNet** benchmark under different non-stationarity levels. The different domains are distinct splits of the original Tiered-ImageNet dataset (please see experimental setup in Section 5 for details on how these splits are obtained).

### B.2. Sensitivity to thresholds $\ell$ and $\tau$ and temperature $\delta$

Figures 3 and 4 illustrate the sensitivity of our algorithm LEEDS with respect to the thresholds $\ell$ and $\tau$ and the temperature parameter $\delta$ in the energy-based detection module. As depicted in Figure 3, when the threshold $\ell$ is too small, the algorithm tends to over detect task switches (as indicated by low Recall for $\ell = 0.5$ in the table), which results in inferior performance of the algorithm due to

| Method | non-stationarity $p = 0.9$ | | non-stationarity $p = 0.75$ | |
|---|---|---|---|---|
| | all domains | ood domains | all domains | ood domains |
| LEEDS | **85.12** ±0.91 | **67.48** ±0.97 | **82.22** ±0.32 | **63.68** ±0.36 |
| CMAML++ | 81.14 ±1.05 | 62.39 ±1.00 | 79.74 ±1.07 | 60.70 ±1.12 |
| FOML | 46.40 ±0.61 | 41.73 ±0.73 | 37.46 ±0.27 | 34.13 ±0.31 |
| MAML | 76.25 ±0.63 | 42.70 ±0.68 | 74.87 ±0.42 | 43.84 ±0.45 |
| ANIL | 64.58 ±0.32 | 34.51 ±0.54 | 72.69 ±0.30 | 35.66 ±0.49 |
| MetaOGD | 72.04 ±0.67 | 46.69 ±0.77 | 67.93 ±0.59 | 42.66 ±0.62 |
| BGD | 25.63 ±0.07 | 25.61 ±0.09 | 27.53 ±0.08 | 27.17 ±0.11 |
| MetaBGD | 53.74 ±0.41 | 42.25 ±0.52 | 40.79 ±0.23 | 34.63 ±0.33 |

Table 4: Average accuracy over 10000 online episodes on **Synbols** benchmark under different non-stationarity levels. The "pre-train" domain corresponds to 3 different alphabets from Synbols dataset, "ood1" corresponds to a new (w.r.t. the pre-training one) alphabet from Synbols dataset, and "ood2" contains font classification tasks.

ineffective reuse of task knowledge. On the other hand when $\ell$ is too large, the high misdetection rate (e.g., indicated by low Precision for $\ell = 5$) results in the algorithm mostly fine-tuning the online task model $\phi_t$ with the current task support data. As expected, this results in a failure mode (the algorithm diverges) due to the adversariality of different tasks. We find that values of $\ell$ in the rage $[1.5, 2.3]$ yield the best performance of our algorithm.

Figure 4 (a) shows that larger values of $\tau$, which collapse to updating the meta-model at each step (even for pretraining task distribution), does not substantially improve the performance. This demonstrates the advantage of the distinct meta-update scheme proposed for in- and out-of-distribution tasks, which avoids unnecessary frequent meta-updates for the pretraining tasks and thus allows a more judicious usage of computational budget. Lower values of $\tau$ (e.g. $\tau = 15$) tend to detect all task distributions as the pertaining one, and thus corresponds to eliminating the domain adaptation component of our algorithm. We also find that simply setting the temperature $\delta = 1$ in the energy expression yields the best performance, and large values of $\delta$ also eliminates the effectiveness of the energy-based detection module (Figure 4 (b)). This is in fact in accordance to the finding in [12], which also suggests setting $\delta = 1$.

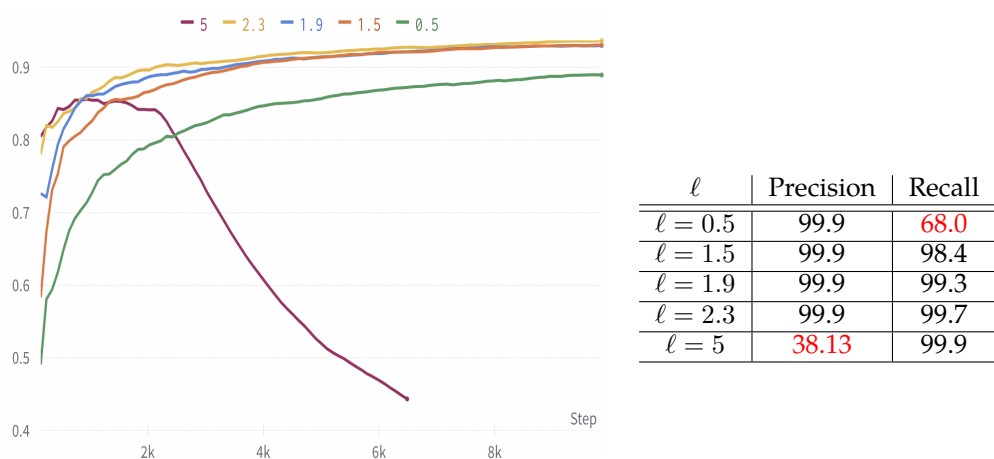

| $\ell$ | Precision | Recall |
|---|---|---|
| $\ell = 0.5$ | 99.9 | 68.0 |
| $\ell = 1.5$ | 99.9 | 98.4 |
| $\ell = 1.9$ | 99.9 | 99.3 |
| $\ell = 2.3$ | 99.9 | 99.7 |
| $\ell = 5$ | 38.13 | 99.9 |

Figure 3: Performance of LEEDS for different values of the threshold $\ell$. **Left plot:** Performance on all encountered domains during online learning. **Right table:** Task boundaries detection for different values of $\ell$. Experiments are conducted on the Omniglot-MNIST-FashionMNIST benchmark.

Figures 5-8 show the online evaluation curves of the different methods for for all settings. The different plots further show that our algorithm LEEDS outperforms other baseline algorithms in the OOD domains and at the same time also retains its performance on the pre-training tasks. The

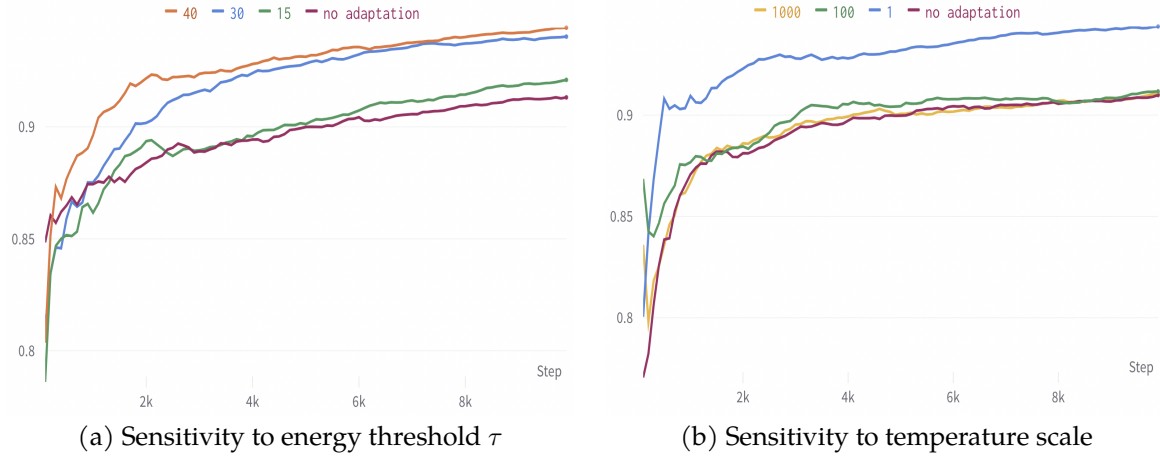

(a) Sensitivity to energy threshold $\tau$        (b) Sensitivity to temperature scale

Figure 4: Performance of LEEDS for: (a) different values of the energy threshold $\tau$ and (b) different scales of the temperature $\delta$. For both plots we report the performance on all encountered domains during online learning. Experiments are conducted on the Omniglot-MNIST-FashionMNIST benchmark.

performance of the methods that do not adapt meta-parameters during online learning phase (such as MAML and ANIL) drops drastically when OOD tasks are far away from the pre-training ones (as shown in Table 1 for **Omniglot-MNIST-FashionMNIST**). For settings in which OOD tasks are close to the pre-training ones (such as in the **Tiered-ImageNet** dataset), MAML can perform similarly to CMAML++.

The superior performance of LEEDS even for the pre-training domain particularly shows that the re-use of task knowledge is beneficial for online meta-learning, as opposed to the usual practice of "resetting" to meta-parameters at each step.

By comparing the two plots in Figure 10, it can be seen that the advantage of our domain adaptation module is more significant when the OOD domains are far away from the pre-training one, as is the case for the FashionMNIST OOD domain compared to the Omniglot pre-training domain.

# C. Further Descriptions about Datasets and Baseline Methods

## C.1. Datasets

We study dynamic online meta-learning on the following benchmarks:

**Omniglot-MNIST-FashionMNIST dataset.** For this dataset, we consider 10-ways 5-shots classification tasks. We pre-train the meta model on a subset of the Omniglot dataset and then deploy it in the online learning environment where tasks are sampled from either the full Omniglot dataset, or from one of the OOD datasets, i.e., the MNIST or FashionMNIST datasets.

**Tiered-ImageNet dataset.** We consider 5-ways 5-shots classification tasks for this dataset. Following [cite paper], we split the original Tiered-ImageNet dataset into the pre-training domain and the OOD domains. More specifically, we pre-train on the first 200 classes of the original training set, use the remaining classes as the first OOD domain (ood1), and set the original test classes as second OOD domain (ood2). During online learning phase, we set the full training set to be IND domain, and ood1 and ood2 as OOD domains. We evaluate algorithms over 20000 online learning episodes.

**Synbols dataset.** We consider 4-ways 4-shots classification tasks in this dataset. The meta model is pre-trained on characters from 3 different alphabets and deployed on characters from a new alphabet (ood1). We also consider font classification tasks as additional OOD tasks (ood2).

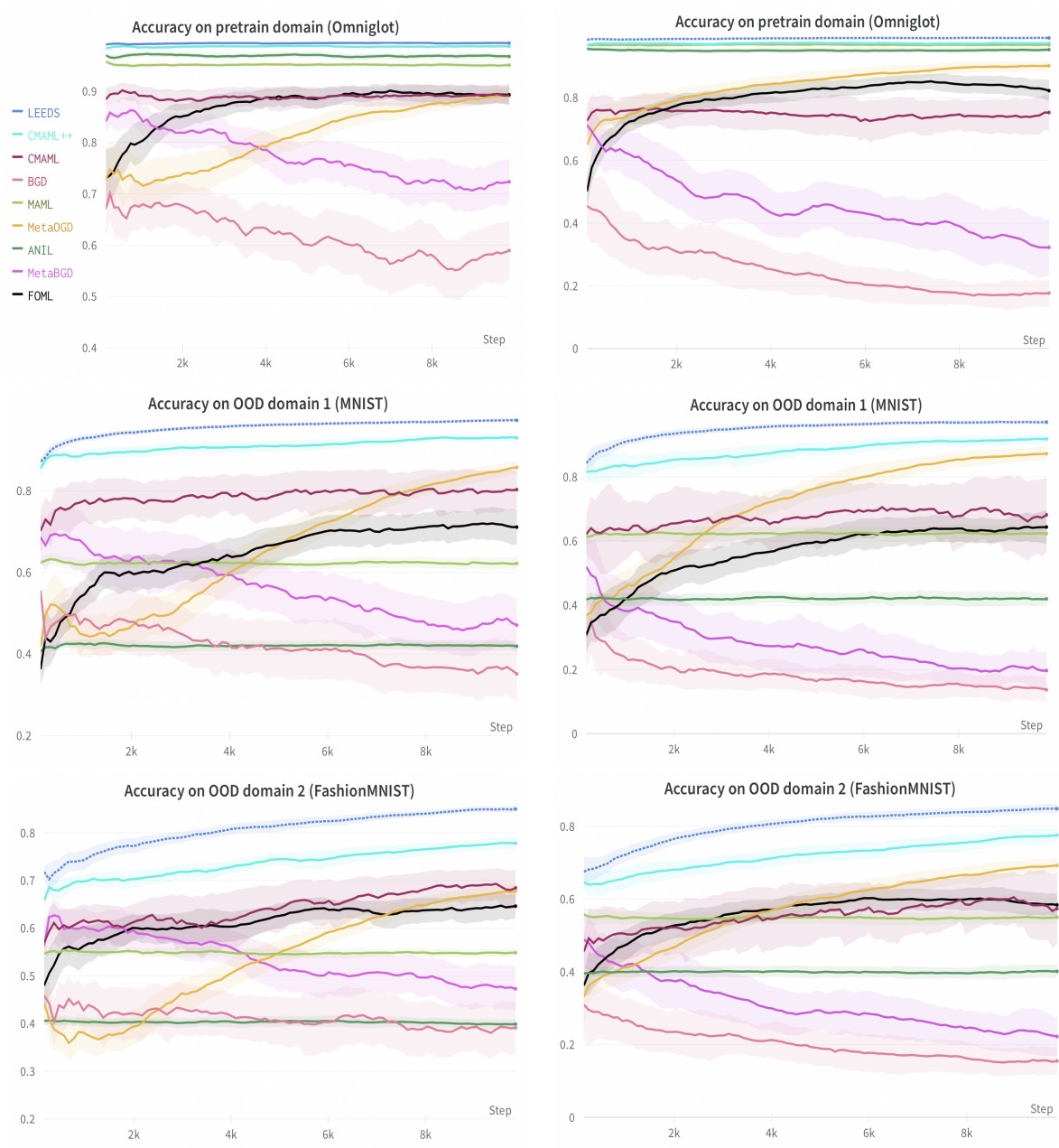

Figure 5: Online evaluations in each of the encountered domains during online learning phase for the **Omniglot-MNIST-FashionMNIST** benchmark. First column corresponds to non-stationarity level $p = 0.9$. In second column $p = 0.75$. LEEDS is the only method that is able to preserve pre-training knowledge while substantially increasing performance in OOD domains. Legend in first plot only.

## C.2. Baseline Methods

We compare our algorithm with the following baseline methods for online meta-learning.

(1) **C-MAML [11] and C-MAML++:** the continual MAML approach (C-MAML) pre-trains the meta model using MAML and employs an online learning strategy based on task boundaries detection. Since C-MAML does not evaluate the task models on separate query sets, for a fair comparison we adapt it to do so and call the resulting algorithm C-MAML++.

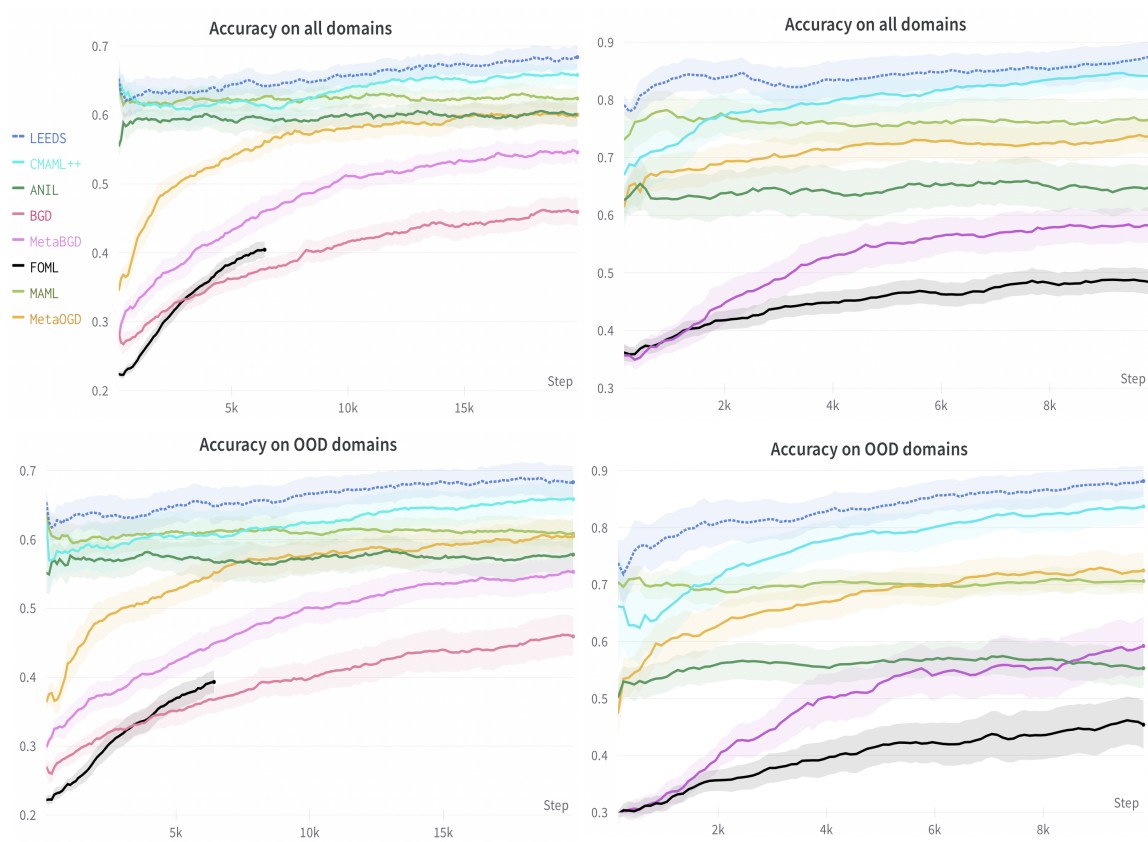

Figure 6: Online evaluations for the **Tiered-ImageNet (TI)** and **Synbols (SB)** benchmarks under $p = 0.9$. First columns correspond to **TI** and second column to **SB**. More results including performance in each domain and under different $p$ can be found in Appendix B. Legend shown in first plot only.

(2) **FOML [10]:** the fully online meta-learning method updates online parameters using the latest online data and maintains a concurrent meta-training process to guide the online updates regularized by the meta model.

(3) **MAML [3] and ANIL [18]:** the MAML baseline consists of an offline pre-training phase and an online deployment phase. During offline pre-training the method learns a common meta-initialization that will be used for all tasks, and the meta-initialization will never be updated at the online stage. The task model is adapted from the meta-initialization using the support set and evaluated on the query set. ANIL is similar to MAML but with partial parameter adaption, i.e., only the last layer is adapted for each task.

(4) **MetaBGD [11] and BGD [48]:** the baseline MetaBGD combines MAML and the Bayesian gradient descent method during online learning.

(5) **MetaOGD [32]:** the meta online gradient descent method simply updates the meta-model at each step using a MAML-like meta-objective evaluated on the current task data.

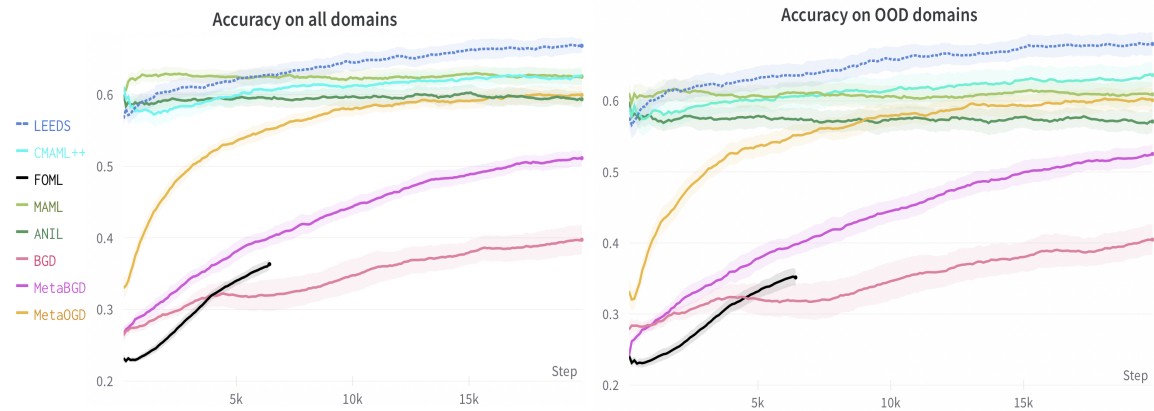

Figure 7: Online evaluations for the **Tiered-ImageNet (TI)** benchmark under **p = 0.75**. **Left:** Accuracies on all encountered domains during online learning. **Right:** Accuracies on all encountered OOD domains during online learning. We compare all baselines on a 16GB GPU memory budget and FOML runs out of memory for this benchmark due to its linear growth in memory requirement.

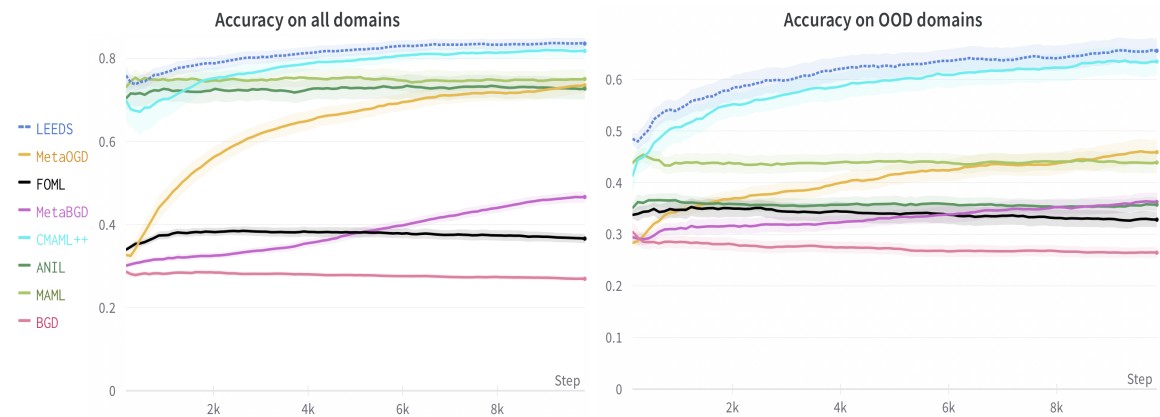

Figure 8: Online evaluations for the **Synbols (SB)** benchmark under **p = 0.75**. **Left:** Accuracies on all encountered domains during online learning. **Right:** Accuracies on all encountered OOD domains during online learning.

# D. Further Experimental Details and Hyperameter Search

## D.1. Further Experimental Specifications

In all our experiments, we consider classification tasks. The cross-entropy loss between predictions and true labels is used to train all models. We use the same convolutional neural network (CNN) architectures widely adopted in few-shot learning literature [3, 11, 13], which include four convolutional blocks followed by a linear classification layer. Each convolutional block is a stack of one $3 \times 3$ convolution layer followed by BatchNormalization, ReLU, and $2 \times 2$ MaxPooling layers. For the **Omniglot-MNIST-FashionMNIST** benchmark, we use 64 filters in each convolutional layer and downsample the gray-scale images to $28 \times 28$ spacial resolution so as to have 64-dimensional feature vectors before classification. For **Tiered-ImageNet** and **Synbols** datasets, the inputs are respectively $3 \times 64 \times 64$ and $3 \times 32 \times 32$ RGB images, resulting respectively in 1024- and 256-dimensional feature vectors for 64 filters in each convolution layer.

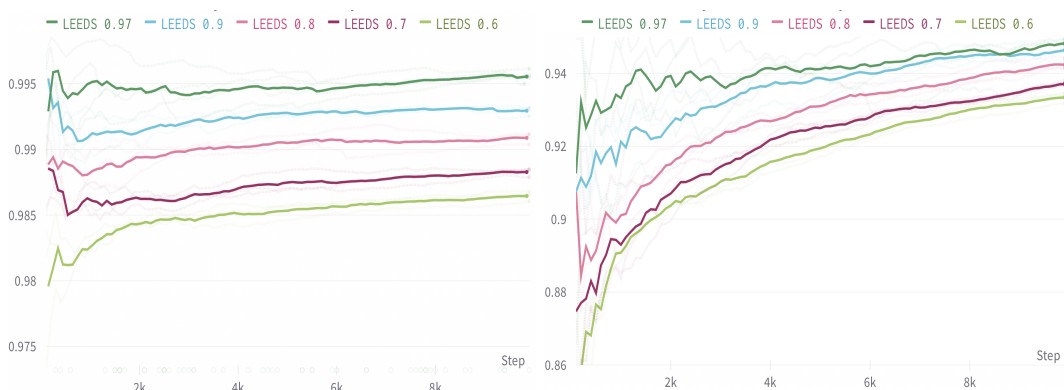

Figure 9: Performance of our algorithm LEEDS under different $p$. **Left plot:** Evaluations in pre-training domain. **Right plot:** Evaluations in all domains.

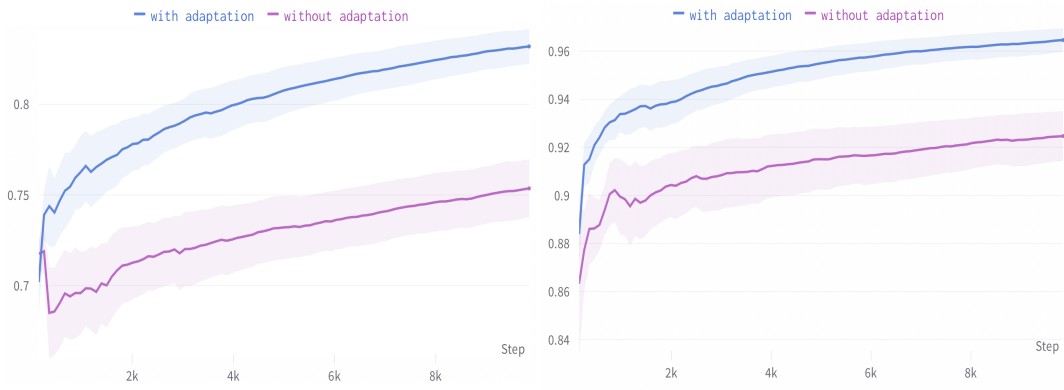

Figure 10: Performance of LEEDS with and without energy-based domain adaptation module. **Left plot:** Evaluations in OOD domain **FashionMINIST**. **Right plot:** Evaluations in OOD domain **MNIST**.

## D.2. Further Implementation Specifications

The implementation of FOML [10] method is not released yet, and hence we compare with our own implementation of their algorithm. For all other baselines, we used their publicly available implementations. Codes for our algorithm LEEDS and all other baselines are provided in the supplementary materials of our submission. All codes are tested with Python 3.6 and Pytorch 1.2.

For example to run our algorithm LEEDS with the best hyperparameters that we obtained for the Omniglot-MNIST-FashionMNIST dataset under $p = 0.9$, one can run the following command:

```
python main.py --algo leeds --use_best 1
```

The experiment setting (e.g., the dataset to use) can be changed in configurations.yaml file. We run all methods on a single NVIDIA Tesla P100 GPU. All compared algorithms except FOML were able to run on 16GB GPU memory. FOML requires at least 32GB to reach 12000 online episodes for the Tiered-ImageNet dataset.

## E. Heuristics for setting the thresholds

For the energy threshold $\tau$, we follow the strategy in Liu et al. (2020), i.e., we set the threshold $\tau$ using the pre-training tasks. More specifically, we set $\tau$ so that $95\%$ of the pre-training inputs are correctly

detected as pre-training data. In standard few-shot learning experimental setups, the true labeling for each individual task is usually randomly chosen. When there is a task switch, the task specific model learnt from the previous task generally does not fit the new task anymore, where the learning performance could be similar to that of a random model. Thus motivated, we find that a good heuristic to choose a starting value for the threshold $\ell$ is to use the loss value evaluated on a random model. For example, for 10-ways classification tasks that value would be $\ell_r = -\log(1/10) = 2.3$.

# F.  Proof of Theorem 4.5

Recall the task-averaged regret:

$$\mathbf{R}_{\text{avg}} = \frac{1}{T} \sum_{t=1}^{T} \left( \sum_{k=1}^{K_t} f_t^k \left( \phi_t^k \right) - \sum_{k=1}^{K_t} f_t^k \left( \phi_t^* \right) \right).$$

We consider the setting where at each round the agent can perform some task-specific adaptation before evaluation on the loss function $f_t^k$ [13]. Let $U_t^k$ be the mapping that defines the adaptation procedure at each round. For instance, a popular choice of the mapping function $U_t^k$ is the one step gradient descent mapping: $U_t^k(\phi) = \phi - \alpha \nabla \hat{f}_t^k(\phi)$, where $\alpha$ is a learning rate and $\hat{f}_t^k$ is an approximation of the loss function $f_t^k$, e.g., computed on a small support set from task $t$.

Denote $\hat{f}_r(\phi_{r-1}) = \frac{1}{|\mathcal{S}_r|} \sum_{\xi \in \mathcal{S}_r} \ell(\phi_{r-1}, \xi) := \hat{f}_{r,r-1}$ as the adaptation loss at round $r$, where $\ell(\cdot, \xi)$ is a classification loss on data point $\xi$. We first provide an upper bound on the detection error.

The total probability of error of the threshold-based detection scheme at round $r$ is given by:

$$P_{\text{error}} = P\left( P_r \neq P_{r-1} \right) P\left( \hat{f}_{r,r-1} < \tau \,\middle|\, P_r \neq P_{r-1} \right) + P\left( P_r = P_{r-1} \right) P\left( \hat{f}_{r,r-1} > \tau \,\middle|\, P_r = P_{r-1} \right) \tag{6}$$

where the first term characterizes the probability of miss detection of the task boundary, and the second term is the probability of false alarm when the underlying task does not change.

Based on Assumption 4.4, for a data distribution $P_r$ and support set $\mathcal{S}_r$ drawn i.i.d. from $P_r$, the Hoeffding inequality yields:

$$P\left( \frac{1}{|\mathcal{S}_r|} \sum_{\xi \in \mathcal{S}_r} \ell(\phi_{r-1}, \xi) - E_{\xi \sim P_r} \ell(\phi_{r-1}, \xi) < -\epsilon \right) \leq \exp\left( \frac{-2|\mathcal{S}_r|\epsilon^2}{M^2} \right)$$

Hence,

$$P\left( \hat{f}_{r,r-1} - \ell_p < -\epsilon \,\middle|\, P_r \neq P_{r-1} \right)$$

$$\leq P\left( \frac{1}{|\mathcal{S}_r|} \sum_{\xi \in \mathcal{S}_r} \ell(\phi_{r-1}, \xi) - E_{\xi \sim P_r} \ell(\phi_{r-1}, \xi) < -\epsilon \,\middle|\, P_r \neq P_{r-1} \right)$$

$$\leq \exp\left( \frac{-2|\mathcal{S}_r|\epsilon^2}{M^2} \right).$$

By setting $\epsilon = \frac{\ell_p - \ell_m}{2}$, we can have

$$P\left( \hat{f}_{r,r-1} < \frac{\ell_m + \ell_p}{2} \,\middle|\, P_r \neq P_{r-1} \right) \leq \exp\left( \frac{-|\mathcal{S}_r|(\ell_p - \ell_m)^2}{2M^2} \right).$$

Thus, setting the threshold $\tau = \frac{\ell_m + \ell_p}{2}$ yields

$$P\left(\hat{f}_{r,r-1} < \tau \,\Big|\, P_r \neq P_{r-1}\right) \leq \exp\left(\frac{-|\mathcal{S}_r|(\ell_p - \ell_m)^2}{2M^2}\right). \tag{7}$$

Using the other side of the Hoeffding inequality, we have:

$$P\left(\hat{f}_{r,r-1} - \ell_m > \epsilon \,\Big|\, P_r = P_{r-1}\right)$$

$$\leq P\left(\frac{1}{|\mathcal{S}_r|}\sum_{\xi \in \mathcal{S}_r} \ell(\phi_{r-1}, \xi) - E_{\xi \sim P_r}\ell(\phi_{r-1}, \xi) > \epsilon \,\Big|\, P_r = P_{r-1}\right)$$

$$\leq P\left(\frac{1}{|\mathcal{S}_r|}\sum_{\xi \in \mathcal{S}_r} \ell(\phi_{r-1}, \xi) - E_{\xi \sim P_r}\ell(\phi_{r-1}, \xi) > \epsilon\right)$$

$$\leq \exp\left(\frac{-2|\mathcal{S}_r|\epsilon^2}{M^2}\right).$$

Therefore, we have:

$$P\left(\hat{f}_{r,r-1} > \frac{\ell_m + \ell_p}{2} \,\Big|\, P_r = P_{r-1}\right) \leq \exp\left(\frac{-|\mathcal{S}_r|(\ell_p - \ell_m)^2}{2M^2}\right),$$

because $\epsilon = \frac{\ell_p - \ell_m}{2}$. Thus, we obtain

$$P\left(\hat{f}_{r,r-1} > \tau \,\Big|\, P_r = P_{r-1}\right) \leq \exp\left(\frac{-|\mathcal{S}_r|(\ell_p - \ell_m)^2}{2M^2}\right). \tag{8}$$

Combining Equations (6), (7) and (8), and using the fact that $P(P_r \neq P_{r-1}) = 1 - P(P_r = P_{r-1})$ can yield the following upper bound on the probability of error when the threshold is set to $\tau = \frac{\ell_m + \ell_p}{2}$:

$$P_{\text{error}} \leq \exp\left(\frac{-|\mathcal{S}_r|(\ell_p - \ell_m)^2}{2M^2}\right). \tag{9}$$

Based on Assumption 4.3, we have for any $\phi$:

$$f_t^k(\phi_t^k) \leq f_t^k(\phi) + \left\langle \nabla f_t^k(\phi), \phi_t^k - \phi \right\rangle + \frac{L}{2}\left\|\phi_t^k - \phi\right\|^2.$$

Therefore, by setting $\phi$ to be the comparator $\phi_t^*$ and using Assumption 4.1, we obtain:

$$f_t^k(\phi_t^k) - f_t^k(\phi_t^*) \leq \frac{L}{2}\left\|\phi_t^k - \phi_t^*\right\|^2.$$

Thus, summing over $k = 1, ..., K_t$ yields

$$\sum_{k=1}^{K_t} f_t^k(\phi_t^k) - \sum_{k=1}^{K_t} f_t^k(\phi_t^*) \leq \frac{L}{2}\sum_{k=1}^{K_t}\left\|\phi_t^k - \phi_t^*\right\|^2. \tag{10}$$

We next bound the term $\sum_{k=1}^{K_t}\left\|\phi_t^k - \phi_t^*\right\|^2$ from above. It follows that

$$\left\|\phi_t^k - \phi_t^*\right\|^2 = \left\|U_t^k(\phi_t^{k-1}) - U_t^k(\phi_t^*)\right\|^2$$

$$\leq \rho^2\left\|\phi_t^{k-1} - \phi_t^*\right\|^2$$

$$\leq \rho^{2k}\left\|\phi_t^0 - \phi_t^*\right\|^2, \tag{11}$$

where the first inequality uses Assumption 4.2. Therefore, combining Equations (10) and (11) gives that

$$\sum_{k=1}^{K_t} f_t^k(\phi_t^k) - \sum_{k=1}^{K_t} f_t^k(\phi_t^*) \leq \frac{L}{2}\sum_{k=1}^{K_t} \rho^{2k}\left\|\phi_t^0 - \phi_t^*\right\|^2$$

$$\leq \frac{L}{2(1-\rho^2)}\left\|\phi_t^0 - \phi_t^*\right\|^2. \tag{12}$$

Hence, summing Equation (12) over $t = 1, ..., T$ yields

$$\sum_{t=1}^{T} \left( \sum_{k=1}^{K_t} f_t^k \left( \phi_t^k \right) - \sum_{k=1}^{K_t} f_t^k \left( \phi_t^* \right) \right) \leq \frac{L}{2(1-\rho^2)} \sum_{t=1}^{T} \left\| \phi_t^0 - \phi_t^* \right\|^2.$$

The task-averaged regret is therefore upper bounded as follows

$$\mathbf{R}_{\mathrm{avg}} \leq \frac{L}{2(1-\rho^2)T} \sum_{t=1}^{T} \left\| \phi_t^0 - \phi_t^* \right\|^2. \tag{13}$$

.

Next, we show that the term $\frac{1}{T} \sum_{t=1}^{T} \left\| \phi_t^0 - \phi_t^* \right\|^2$ converges as $T \to \infty$. We have,

$$\frac{1}{T} \sum_{t=1}^{T} \left\| \phi_t^0 - \phi_t^* \right\|^2 = \frac{1}{T} \sum_{t=1}^{T} \left( \left\| \phi_t^0 - \phi_t^* \right\|^2 - \left\| \phi_t^* - \phi^* \right\|^2 \right) + \frac{1}{T} \sum_{t=1}^{T} \left\| \phi_t^* - \phi^* \right\|^2, \tag{14}$$

where $\phi_* = \frac{1}{T} \sum_{t=1}^{T} \phi_t^*$ is the mean of the dynamic comparators. Note that $\phi^*$ is the minimizer of the summation $\sum_{t=1}^{T} \left\| \phi_t^* - \phi^* \right\|^2$. Thus, if we use an algorithm such as OGD [32] to update the initialization $\phi_t^0$ with loss $\left\| \phi_t^0 - \phi_t^* \right\|^2$ then summation in the first term at the right hand side of Equation (14) corresponds to the regret of OGD on a stream of strongly-convex functions, which is well-known to be $\mathcal{O} \left( \log T \right)$. The second term $\frac{1}{T} \sum_{t=1}^{T} \left\| \phi_t^* - \phi^* \right\|^2 = \sigma_*^2$ corresponds to the variance of the dynamic comparators. Hence, combining Equations (13) and (14) we obtain

$$\mathbf{R}_{\mathrm{avg}} = \mathcal{O} \left( \frac{\sigma_*^2 + \frac{\log T}{T}}{1 - \rho^2} \right). \tag{15}$$

Using Equation (9) for $R = \sum_{t=1}^{T} K_t$ rounds, we have with probability at least $1 - R \exp \left( \frac{-S(\ell_p - \ell_m)^2}{2M^2} \right)$ the task-average regret is bounded as in Equation (15). Hence, the expected task-averaged regret for the algorithm without exact task-boundary detection is upper bounded as

$$\mathbb{E} \, \mathbf{R}_{\mathrm{avg}} \leq \mathcal{O} \left( \frac{\sigma_*^2 + \frac{\log T}{T}}{1 - \rho^2} + R^2 \exp \left( \frac{-S(\ell_p - \ell_m)^2}{2M^2} \right) \right) \leq \mathcal{O} \left( \frac{\sigma_*^2 + \frac{\log T}{T}}{1 - \rho^2} + R^{-\left( \frac{c(\ell_p - \ell_m)^2}{2M^2} - 2 \right)} \right), \tag{16}$$

where the second inequality follows by selecting support sets of size $S = c \log R$, which completes the proof of Theorem 4.5. Further choosing $c > \frac{4M^2}{(\ell_p - \ell_m)^2}$ ensures a constant expected task-averaged regret.

