# OpenReview forum: "Algorithm Design for Online Meta-Learning with Task Boundary Detection"
_CPAL.cc/2024/Conference — CPAL 2024 (Proceedings Track) Oral_

### Meta-Review · Area_Chair_vZNt · 2023-11-13

**Recommendation:** Accept (Oral)
**Confidence:** 4

**Metareview:**

(This is an invited paper, so only one meta-review is provided)

This work proposed a new, task-agnostic online meta-learning algorithm for dynamic environments. It addresses key challenges in the field by allowing for the algorithm to operate without fixed task distributions or known task boundaries. The introduction of mechanisms to detect task switches and distribution shifts based on empirical observations is particularly commendable. These mechanisms are crucial for the algorithm's adaptability, enabling it to maintain performance on in-distribution tasks while swiftly adapting to new, out-of-distribution tasks. The avoidance of data storage from previous tasks and the demonstration of sublinear task-averaged regret are also impressive features that suggest significant improvements over existing approaches. Empirical validation across multiple benchmarks substantiates the algorithm's efficacy. Therefore, the paper appears to be a valuable contribution and is recommended for acceptance.

---

### Decision · Program_Chairs · 2023-11-20

Accept (Oral)